# Scalable Inference for Neuronal Connectivity from Calcium Imaging

Alyson K. Fletcher                    Sundeep Rangan

## Abstract

Fluorescent calcium imaging provides a potentially powerful tool for inferring connectivity in neural circuits with up to thousands of neurons. However, a key challenge in using calcium imaging for connectivity detection is that current systems often have a temporal response and frame rate that can be orders of magnitude slower than the underlying neural spiking process. Bayesian inference methods based on expectation-maximization (EM) have been proposed to overcome these limitations, but are often computationally demanding since the E-step in the EM procedure typically involves state estimation for a high-dimensional nonlinear dynamical system. In this work, we propose a computationally fast method for the state estimation based on a hybrid of loopy belief propagation and approximate message passing (AMP). The key insight is that a neural system as viewed through calcium imaging can be factorized into simple scalar dynamical systems for each neuron with linear interconnections between the neurons. Using the structure, the updates in the proposed hybrid AMP methodology can be computed by a set of one-dimensional state estimation procedures and linear transforms with the connectivity matrix. This yields a computationally scalable method for inferring connectivity of large neural circuits. Simulations of the method on realistic neural networks demonstrate good accuracy with computation times that are potentially significantly faster than current approaches based on Markov Chain Monte Carlo methods.

## 1 Introduction

Determining connectivity in populations of neurons is fundamental to understanding neural computation and function. In recent years, calcium imaging has emerged as a promising technique for measuring synaptic activity and mapping neural micro-circuits [1–4]. Fluorescent calcium-sensitive dyes and genetically-encoded calcium indicators can be loaded into neurons, which can then be imaged for spiking activity either *in vivo* or *in vitro*. Current methods enable imaging populations of hundreds to thousands of neurons with very high spatial resolution. Using two-photon microscopy, imaging can also be localized to specific depths and cortical layers [5]. Calcium imaging also has the potential to be combined with optogenetic stimulation techniques such as in [6].

However, inferring neural connectivity from calcium imaging remains a mathematically and computationally challenging problem. Unlike anatomical methods, calcium imaging does not directly measure connections. Instead, connections must be inferred indirectly from statistical relationships between spike activities of different neurons. In addition, the measurements of the spikes from calcium imaging are indirect and noisy. Most importantly, the imaging introduces significant temporal blurring of the spike times: the typical time constants for the decay of the fluorescent calcium concentration, $[Ca^{2+}]$, can be on the order of a second – orders of magnitude slower than the spike rates and inter-neuron dynamics. Moreover, the calcium imaging frame rate remains relatively slow – often less than 100 Hz. Hence, determining connectivity typically requires super-resolution of spike times within the frame period.

To overcome these challenges, the recent work [7] proposed a Bayesian inference method to estimate functional connectivity from calcium imaging in a systematic manner. Unlike "model-free" approaches such as in [8], the method in [7] assumed a detailed functional model of the neural dynamics with unknown parameters including a connectivity weight matrix $\mathbf{W}$. The model parameters including the connectivity matrix can then be estimated via a standard EM procedure [9]. While the method is general, one of the challenges in implementing it is the computational complexity. As we discuss below, the E-step in the EM procedure essentially requires estimating the distributions of hidden states in a nonlinear dynamical system whose state dimension grows linearly with the number of neurons. Since exact computation of these densities grows exponentially in the state dimension, [7] uses an approximate method based on blockwise Gibbs sampling where each block of variables consists of the hidden states associated with one neuron. Since the variables within a block are described as a low-dimensional dynamical system, the updates of the densities for the Gibbs sampling can be computed efficiently via a standard particle filter [10, 11]. However, simulations of the method show that the mixing between blocks can still take considerable time to converge.

This paper provides a novel method that can potentially significantly improve the computation time of the state estimation. The key insight is to recognize that a high-dimensional neural system can be "factorized" into simple, scalar dynamical systems for each neuron with linear interactions between the neurons. As described below, we assume a standard leaky integrate-and-fire model for each neuron [12] and a first-order AR process for the calcium imaging [13]. Under this model, the dynamics of $N$ neurons can be described by $2N$ systems, each with a scalar (i.e. one-dimensional) state. The coupling between the systems will be linear as described by the connectivity matrix $\mathbf{W}$. Using this factorization, approximate state estimation can then be efficiently performed via approximations of loopy belief propagation (BP) [14]. Specifically, we show that the loopy BP updates at each of the factor nodes associated with the integrate-and-fire and calcium imaging can be performed via a scalar standard forward–backward filter. For the updates associated with the linear transform $\mathbf{W}$, we use recently-developed approximate message passing (AMP) methods.

AMP was originally proposed in [15] for problems in compressed sensing. Similar to expectation propagation [16], AMP methods use Gaussian and quadratic approximations of loopy BP but with further simplifications that leverage the linear interactions. AMP was used for neural mapping from multi-neuron excitation and neural receptive field estimation in [17, 18]. Here, we use a so-called hybrid AMP technique proposed in [19] that combines AMP updates across the linear coupling terms with standard loopy BP updates on the remainder of the system. When applied to the neural system, we show that the estimation updates become remarkably simple: For a system with $N$ neurons, each iteration involves running $2N$ forward–backward scalar state estimation algorithms, along with multiplications by $\mathbf{W}$ and $\mathbf{W}^T$ at each time step. The practical complexity scales as $O(NT)$ where $T$ is the number of time steps. We demonstrate that the method can be significantly faster than the blockwise Gibbs sampling proposed in [7], with similar accuracy.

## 2 System Model

We consider a recurrent network of $N$ spontaneously firing neurons. All dynamics are approximated in discrete time with some time step $\Delta$, with a typical value $\Delta = 1$ ms. Importantly, this time step is typically smaller than the calcium imaging period, so the model captures the dynamics between observations. Time bins are indexed by $k = 0, \ldots, T-1$, where $T$ is the number of time bins so that $T\Delta$ is the total observation time in seconds. Each neuron $i$ generates a sequence of spikes (action potentials) indicated by random variables $s_i^k$ taking values 0 or 1 to represent whether there was a spike in time bin $k$ or not. It is assumed that the discretization step $\Delta$ is sufficiently small such that there is at most one action potential from a neuron in any one time bin. The spikes are generated via a standard leaky integrate-and-fire (LIF) model [12] where the (single compartment) membrane voltage $v_i^k$ of each neuron $i$ and its corresponding spike output sequence $s_i^k$ evolve as

$$\tilde{v}_i^{k+1} = (1 - \alpha_{IF})v_i^k + q_i^k + d_{v_i}^k, \quad q_i^k = \sum_{j=1}^N W_{ij}s_j^{k-\delta} + b_{IF,i}, \quad d_{v_i}^k \sim \mathcal{N}(0, \tau_{IF}), \quad (1)$$

and

$$(v_i^{k+1}, s_i^{k+1}) = \begin{cases} (\tilde{v}_i^k, 0) & \text{if } v_i^k < \mu, \\ (0, 1) & \text{if } \tilde{v}_i^k \geq \mu, \end{cases} \quad (2)$$

where $\alpha_{IF}$ is a time constant for the integration leakage; $\mu$ is the threshold potential at which the neurons spikes; $b_{IF,i}$ is a constant bias term; $q_i^k$ is the increase in the membrane potential from the pre-synaptic spikes from other neurons and $d_{v_i}^k$ is a noise term including both thermal noise and currents from other neurons that are outside the observation window. The voltage has been scaled so that the reset voltage is zero. The parameter $\delta$ is the integer delay (in units of the time step $\Delta$) between the spike in one neuron and the increase in the membrane voltage in the post-synaptic neuron. An implicit assumption in this model is the post-synaptic current arrives in a single time bin with a fixed delay.

To determine functional connectivity, the key parameter to estimate will be the matrix $\mathbf{W}$ of the weighting terms $W_{ij}$ in (1). Each parameter $W_{ij}$ represents the increase in the membrane voltage in neuron $i$ due to the current triggered from a spike in neuron $j$. The connectivity weight $W_{ij}$ will be zero whenever neuron $j$ has no connection to neuron $i$. Thus, determining $\mathbf{W}$ will determine which neurons are connected to one another and the strengths of those connections.

For the calcium imaging, we use a standard model [7], where the concentration of fluorescent Calcium has a fast initial rise upon an action potential followed by a slow exponential decay. Specifically, we let $z_i^k = [\text{Ca}^{2+}]_k$ be the concentration of fluorescent Calcium in neuron $i$ in time bin $k$ and assume it evolves as first-order auto-regressive $AR(1)$ model,

$$z_i^{k+1} = (1 - \alpha_{CA,i})z_i^k + s_i^k, \tag{3}$$

where $\alpha_{CA}$ is the Calcium time constant. The observed net fluorescence level is then given by a noisy version of $z_i^k$,

$$y_i^k = a_{CA,i}z_i^k + b_{CA,i} + d_{y_i}^k, \quad d_{y_i}^k \sim \mathcal{N}(0, \tau_y), \tag{4}$$

where $a_{CA,i}$ and $b_{CA,i}$ are constants and $d_{y_i}$ is white Gaussian noise with variance $\tau_y$. Nonlinearities such as saturation described in [13] can also be modeled.

As mentioned in the Introduction, a key challenge in calcium imaging is the relatively slow frame rate which has the effect of subsampling of the fluorescence. To model the subsampling, we let $I_F$ denote the set of time indices $k$ on which we observe $F_i^k$. We will assume that fluorescence values are observed once every $T_F$ time steps for some integer period $T_F$ so that $I_F = \{0, T_F, 2T_F, \ldots, KT_F\}$ where $K$ is the number of Calcium image frames.

## 3 Parameter Estimation via Message Passing

### 3.1 Problem Formulation

Let $\theta$ be set of all the unknown parameters,

$$\theta = \{\mathbf{W}, \tau_{IF}, \tau_{CA}, \alpha_{IF}, b_{IF,i}, \alpha_{CA}, a_{CA,i}, b_{CA,i}, i = 1, \ldots, N\}, \tag{5}$$

which includes the connectivity matrix, time constants and various variances and bias terms. Estimating the parameter set $\theta$ will provide an estimate of the connectivity matrix $\mathbf{W}$, which is our main goal.

To estimate $\theta$, we consider a regularized maximum likelihood (ML) estimate

$$\widehat{\theta} = \arg\max_{\theta} L(\mathbf{y}|\theta) + \phi(\theta), \quad L(\mathbf{y}|\theta) = -\log p(\mathbf{y}|\theta), \tag{6}$$

where $\mathbf{y}$ is the set of observed values; $L(\mathbf{y}|\theta)$ is the negative log likelihood of $\mathbf{y}$ given the parameters $\theta$ and $\phi(\theta)$ is some regularization function. For the calcium imaging problem, the observations $\mathbf{y}$ are the observed fluorescence values across all the neurons,

$$\mathbf{y} = \{\mathbf{y}_1, \ldots, \mathbf{y}_N\}, \quad \mathbf{y}_i = \{y_i^k, \quad k \in I_F\}, \tag{7}$$

where $\mathbf{y}_i$ is the set of fluorescence values from neuron $i$, and, as mentioned above, $I_F$ is the set of time indices $k$ on which the fluorescence is sampled.

The regularization function $\phi(\theta)$ can be used to impose constraints or priors on the parameters. In this work, we will assume a simple regularizer that only constrains the connectivity matrix $\mathbf{W}$,

$$\phi(\theta) = \lambda\|\mathbf{W}\|_1, \quad \|\mathbf{W}\|_1 := \sum_{ij} |W_{ij}|, \tag{8}$$

where $\lambda$ is a positive constant. The $\ell_1$ regularizer is a standard convex function used to encourage sparsity [20], which we know in this case must be valid since most neurons are not connected to one another.

## 3.2 EM Estimation

Exact computation of $\widehat{\theta}$ in (6) is generally intractable, since the observed fluorescence values $\mathbf{y}$ depend on the unknown parameters $\theta$ through a large set of hidden variables. Similar to [7], we thus use a standard EM procedure [9]. To apply the EM procedure to the calcium imaging problem, let $\mathbf{x}$ be the set of hidden variables,

$$\mathbf{x} = \{\mathbf{v}, \mathbf{z}, \mathbf{q}, \mathbf{s}\}, \tag{9}$$

where $\mathbf{v}$ are the membrane voltages of the neurons, $\mathbf{z}$ the calcium concentrations, $\mathbf{s}$ the spike outputs and $\mathbf{q}$ the linearly combined spike inputs. For any of these variables, we will use the subscript $i$ (e.g. $\mathbf{v}_i$) to denote the values of the variables of a particular neuron $i$ across all time steps and superscript $k$ (e.g. $\mathbf{v}^k$) to denote the values across all neurons at a particular time step $k$. Thus, for the membrane voltage

$$\mathbf{v} = \{v_i^k\}, \quad \mathbf{v}^k = \left(v_1^k, \ldots, v_N^k\right), \quad \mathbf{v}_i = \left(v_i^0, \ldots, v_i^{T-1}\right).$$

The EM procedure alternately estimates distributions on the hidden variables $\mathbf{x}$ given the current parameter estimate for $\theta$ (the E-step); and then updates the estimates for parameter vector $\theta$ given the current distribution on the hidden variables $\mathbf{x}$ (the M-step).

- *E-Step:* Given parameter estimates $\widehat{\theta}^\ell$, estimate

$$P(\mathbf{x}|\mathbf{y}, \widehat{\theta}^\ell), \tag{10}$$

which is the posterior distribution of the hidden variables $\mathbf{x}$ given the observations $\mathbf{y}$ and current parameter estimate $\widehat{\theta}^\ell$.

- *M-step* Update the parameter estimate via the minimization,

$$\widehat{\theta}^{\ell+1} = \arg\min_\theta \mathbb{E}\left[L(\mathbf{x}, \mathbf{y}|\theta)|\widehat{\theta}^\ell\right] + \phi(\theta), \tag{11}$$

where $L(\mathbf{x}, \mathbf{y}|\theta)$ is the joint negative log likelihood,

$$L(\mathbf{x}, \mathbf{y}|\theta) = -\log p(\mathbf{x}, \mathbf{y}|\theta). \tag{12}$$

In (11) the expectation is with respect to the distribution found in (10) and $\phi(\theta)$ is the parameter regularization function.

The next two sections will describe how we approximately perform each of these steps.

## 3.3 E-Step estimation via Approximate Message Passing

For the calcium imaging problem, the challenging step of the EM procedure is the E-step, since the hidden variables $\mathbf{x}$ to be estimated are the states and outputs of a high-dimensional nonlinear dynamical system. Under the model in Section 2, a system with $N$ neurons will require $N$ states for the membrane voltages $v_i^k$ and $N$ states for the bound Ca concentration levels $z_i^k$, resulting in a total state dimension of $2N$. The E-step for this system is essentially a state estimation problem, and exact inference of the states of a general nonlinear dynamical system grows exponentially in the state dimension. Hence, exact computation of the posterior distribution (10) for the system will be intractable even for a moderately sized network.

As described in the Introduction, we thus use an approximate messaging passing method that exploits the separable structure of the system. For the remainder of this section, we will assume the parameters $\theta$ in (5) are fixed to the current parameter estimate $\widehat{\theta}^\ell$. Then, under the assumptions of Section 2, the joint probability distribution function of the variables can be written in a factorized form,

$$P(\mathbf{x}, \mathbf{y}) = P(\mathbf{q}, \mathbf{v}, \mathbf{s}, \mathbf{z}, \mathbf{y}) = \frac{1}{Z} \prod_{k=0}^{T-1} \mathbb{1}_{\{\mathbf{q}^k = \mathbf{W}\mathbf{s}^k\}} \prod_{i=1}^{N} \psi_i^{IF}(\mathbf{q}_i, \mathbf{v}_i, \mathbf{s}_i) \psi_i^{CA}(\mathbf{s}_i, \mathbf{z}_i, \mathbf{y}_i), \tag{13}$$

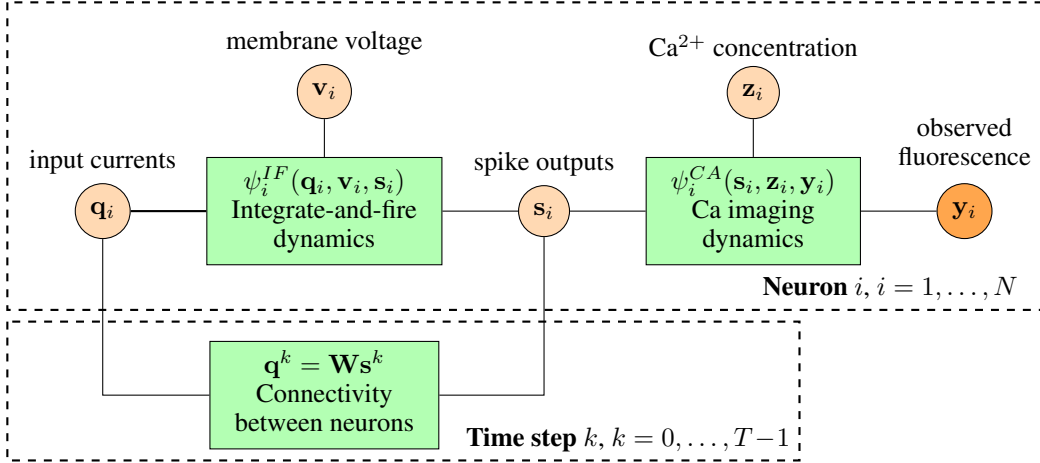

Figure 1: Factor graph plate representation of the system where the spike dynamics are described by the factor node $\psi_i^{IF}(\mathbf{q}_i, \mathbf{v}_i, \mathbf{s}_i)$ and the calcium image dynamics are represented via the factor node $\psi_i^{CA}(\mathbf{s}_i, \mathbf{z}_i, \mathbf{y}_i)$. The high-dimensional dynamical system is described as $2N$ scalar dynamical systems (2 for each neuron) with linear interconnections, $\mathbf{q}^k = \mathbf{W}\mathbf{s}^k$ between the neurons. A computational efficient approximation of loopy BP [19] is applied to this graph for approximate Bayesian inference required in the E-step of the EM algorithm.

where $Z$ is a normalization constant; $\psi_i^{IF}(\mathbf{q}_i, \mathbf{v}_i, \mathbf{s}_i)$ is the potential function relating the summed spike inputs $\mathbf{q}_i$ to the membrane voltages $\mathbf{v}_i$ and spike outputs $\mathbf{s}_i$; $\psi_i^{CA}(\mathbf{s}_i, \mathbf{z}_i, \mathbf{y}_i)$ relates the spike outputs $\mathbf{s}_i$ to the bound calcium concentrations $\mathbf{z}_i$ and observed fluorescence values $\mathbf{y}_i$; and the term $\mathbb{1}_{\{\mathbf{q}^k = \mathbf{W}\mathbf{s}^k\}}$ indicates that the distribution is to be restricted to the set satisfying the linear constraints $\mathbf{q}^k = \mathbf{W}\mathbf{s}^k$ across all time steps $k$.

As in standard loopy BP [14], we represent the distribution (13) in a *factor graph* as shown in Fig. 1. Now, for the E-step, we need to compute the marginals of the posterior distribution $p(\mathbf{x}|\mathbf{y})$ from the joint distribution (13). Using the factor graph representation, loopy BP iteratively updates estimates of these marginal posterior distributions using a message passing procedure, where the estimates of the distributions (called beliefs) are passed between the variable and factor nodes in the graph. In general, the computationally challenging component of loopy BP is the updates on the belief messages at the factor nodes. However, using the factorized structure in Fig. 1 along with approximate message passing (AMP) simplifications as described in [19], these updates can be computed easily.

Details are given in the full paper [21], but the basic procedure for the factor node updates and the reasons why these computations are simple can be summarized as follows. At a high level, the factor graph structure in Fig. 1 partitions the $2N$-dimensional nonlinear dynamical system into $N$ scalar systems associated with each membrane voltage $v_i^k$ and an additional $N$ scalar systems associated with each calcium concentration level $z_i^k$. The only coupling between these systems is through the linear relationships $\mathbf{q}^k = \mathbf{W}\mathbf{s}^k$. As shown in Appendix **??**, on each of the scalar systems, the factor node updates required by loopy BP essentially reduces to a state estimation problem for this system. Since the state space of this system is scalar (i.e. one-dimensional), we can discretize the state space well with a small number of points – in the experiments below we use $L = 20$ points per dimension. Once discretized, the state estimation can be performed via a standard forward–backward algorithm. If there are $T$ time steps, the algorithm will have a computational cost of $O(TL^2)$ per scalar system. Hence, all the factor node updates across all the $2N$ scalar systems has total complexity $O(NTL^2)$.

For the factor nodes associated with the linear constraints $\mathbf{q}^k = \mathbf{W}\mathbf{s}^k$, we use the AMP approximations [19]. In this approximation, the messages for the transform outputs $q_i^k$ are approximated as Gaussians which is, at least heuristically, justified since the they are outputs of a linear transform of a large number of variables, $s_i^k$. In the AMP algorithm, the belief updates for the variables $\mathbf{q}^k$ and $\mathbf{s}^k$ can then be computed simply by linear transformations of $\mathbf{W}$ and $\mathbf{W}^T$. Since $\mathbf{W}$ represents a connectivity matrix, it is generally sparse. If each row of $\mathbf{W}$ has $d$ non-zero values, multiplication

by $\mathbf{W}$ and $\mathbf{W}^T$ will be $O(Nd)$. Performing the multiplications across all time steps results in a total complexity of $O(NTd)$.

Thus, the total complexity of the proposed E-step estimation method is $O(NTL^2 + NTd)$ per loopy BP iteration. We typically use a small number of loopy BP iterations per EM update (in fact, in the experiments below, we found reasonable performance with one loopy BP update per EM update). In summary, we see that while the overall neural system is high-dimensional, it has a linear + scalar structure. Under the assumption of the bounded connectivity $d$, this structure enables an approximate inference strategy that scales linearly with the number of neurons $N$ and time steps $T$. Moreover, the updates in different scalar systems can be computed separately allowing a readily parallelizable implementation.

## 3.4 Approximate M-step Optimization

The M-step (11) is computationally relatively simple. All the parameters in $\theta$ in (5) have a linear relationship between the components of the variables in the vector $\mathbf{x}$ in (9). For example, the parameters $a_{CA,i}$ and $b_{CA,i}$ appear in the fluorescence output equation (4). Since the noise $d_{y_i}^k$ in this equation is Gaussian, the negative log likelihood (12) is given by

$$L(\mathbf{x}, \mathbf{y}|\theta) = \frac{1}{2\tau_{y_i}} \sum_{k \in I_F} (y_i^k - a_{CA,i} z_i^k - b_{CA,i})^2 + \frac{T}{2} \log(\tau_{y_i}) + \text{other terms},$$

where "other terms" depend on parameters other than $a_{CA,i}$ and $b_{CA,i}$. The expectation $\mathbb{E}(L(\mathbf{x}, \mathbf{y}|\theta)|\widehat{\theta}^\ell)$ will then depend only on the mean and variance of the variables $y_i^k$ and $z_i^k$, which are provided by the E-step estimation. Thus, the M-step optimization in (11) can be computed via a simple least-squares problem. Using the linear relation (1), a similar method can be used for $\alpha_{IF,i}$ and $b_{IF,i}$, and the linear relation (3) can be used to estimate the calcium time constant $\alpha_{CA}$.

To estimate the connectivity matrix $\mathbf{W}$, let $\mathbf{r}^k = \mathbf{q}^k - \mathbf{W}\mathbf{s}^k$ so that the constraints in (13) is equivalent to the condition that $\mathbf{r}^k = 0$. Thus, the term containing $\mathbf{W}$ in the expectation of the negative log likelihood $\mathbb{E}(L(\mathbf{x}, \mathbf{y}|\theta)|\widehat{\theta}^\ell)$ is given by the negative log probability density of $\mathbf{r}^k$ evaluated at zero. In general, this density will be a complex function of $\mathbf{W}$ and difficult to minimize. So, we approximate the density as follows: Let $\widehat{\mathbf{q}}$ and $\widehat{\mathbf{s}}$ be the expectation of the variables $\mathbf{q}$ and $\mathbf{s}$ given by the E-step. Hence, the expectation of $\mathbf{r}^k$ is $\widehat{\mathbf{q}}^k - \mathbf{W}\widehat{\mathbf{s}}^k$. As a simple approximation, we will then assume that the variables $r_i^k$ are Gaussian, independent and having some constant variance $\sigma^2$. Under this simplifying assumption, the M-step optimization of $\mathbf{W}$ with the $\ell_1$ regularizer (8) reduces to

$$\widehat{\mathbf{W}} = \arg\min_{\mathbf{W}} \frac{1}{2} \sum_{k=0}^{T-1} \|\widehat{\mathbf{q}}^k - \mathbf{W}\widehat{\mathbf{s}}^k\|^2 + \sigma^2 \lambda \|\mathbf{W}\|_1, \qquad (14)$$

For a given value of $\sigma^2 \lambda$, the optimization (14) is a standard LASSO optimization [22] which can be evaluated efficiently via a number of convex programming methods. In this work, in each M-step, we adjust the regularization parameter $\sigma^2 \lambda$ to obtain a desired fixed sparsity level in the solution $\mathbf{W}$.

## 3.5 Initial Estimation via Sparse Regression

Since the EM algorithm cannot be guaranteed to converge a global maxima, it is important to pick the initial parameter estimates carefully. The time constants and noise levels for the calcium image can be extracted from the second-order statistics of fluorescence values and simple thresholding can provide a coarse estimate of the spike rate.

The key challenge is to obtain a good estimate for the connectivity matrix $\mathbf{W}$. For each neuron $i$, we first make an initial estimate of the spike probabilities $P(s_i^k = 1|\mathbf{y}_i)$ from the observed fluorescence values $\mathbf{y}_i$, assuming some i.i.d. prior of the form $P(s_i^t) = \lambda\Delta$, where $\lambda$ is the estimated average spike rate per second. This estimation can be solved with the filtering method in [13] and is also equivalent to the method we use for the factor node updates. We can then threshold these probabilities to make a hard initial decision on each spike: $s_i^k = 0$ or 1. We then propose to estimate $\mathbf{W}$ from the spikes as follows. Fix a neuron $i$ and let $\mathbf{w}_i$ be the vector of weights $W_{ij}$, $j = 1, \ldots, N$. Under the assumption that the initial spike sequence $s_i^k$ is exactly correct, it is shown in the full paper [21], that

| Parameter | Value |
|---|---|
| Number of neurons, $N$ | 100 |
| Connection sparsity | 10% with random connections. All connections are excitatory with the non-zero weights $W_{ij}$ being exponentially distributed. |
| Mean firing rate per neuron | 10 Hz |
| Simulation time step, $\Delta$ | 1 ms |
| Total simulation time, $T\Delta$ | 10 sec (10,000 time steps) |
| Integration time constant, $\alpha_{IF}$ | 20 ms |
| Conduction delay, $\delta$ | 2 time steps = 2 ms |
| Integration noise, $d_{v_i}^k$ | Produced from two unobserved neurons. |
| Ca time constant, $\alpha_{CA}$ | 500 ms |
| Fluorescence noise, $\tau_{CA}$ | Set to 20 dB SNR |
| Ca frame rate , $1/T_F$ | 100 Hz |

Table 1: Parameters for the Ca image simulation.

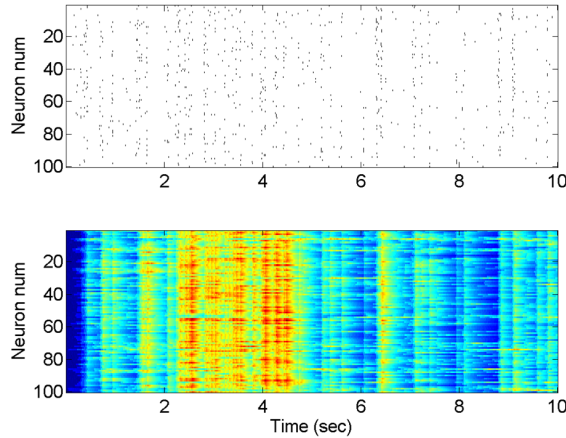

Figure 2: Typical network simulation trace. Top panel: Spike traces for the 100 neuron simulated network. Bottom panel: Calcium image fluorescence levels. Due to the random network topology, neurons often fire together, significantly complicating connectivity detection. Also, as seen in the lower panel, the slow decay of the fluorescent calcium blurs the spikes in the calcium image.

a regularized maximum likelihood estimate of $\mathbf{w}_i$ and bias term $b_{IF,i}$ is given by

$$(\widehat{\mathbf{w}}_i, \widehat{b}_{IF,i}) = \arg\min_{\mathbf{w}_i, b_{IF,i}} \sum_{k=0}^{T-1} L_{ik}(\mathbf{u}_k^T \mathbf{w}_i + c_{ik} b_{IF,i} - \mu, s_i^k) + \lambda \sum_{j=1}^{N} |W_{ij}|, \qquad (15)$$

where $L_{ik}$ is a probit loss function and the vector $\mathbf{u}_k$ and scalar $c_{ik}$ can be determined from the spike estimates. The optimization (15) is precisely a standard probit regression used in sparse linear classification [23]. This form arises due to the nature of the leaky integrate-and-fire model (1) and (2). Thus, assuming the initial spike sequences are estimated reasonably accurately, one can obtain good initial estimates for the weights $W_{ij}$ and bias terms $b_{IF,i}$ by solving a standard classification problem.

## 4  Numerical Example

The method was tested using realistic network parameters, as shown in Table 1, similar to those found in neurons networks within a cortical column [24]. Similar parameters are used in [7]. The network consisted of 100 neurons with each neuron randomly connected to 10% of the other neurons. The non-zero weights $W_{ij}$ were drawn from an exponential distribution. As a simplification, all weights were positive (i.e. the neurons were excitatory – there were no inhibitory neurons in the simulation). A typical random matrix $\mathbf{W}$ generated in this manner would not in general result in a stable system. To stabilize the system, we followed the procedure in [8] where the system is simulated multiple times. After each simulation, the rows of the matrix $\mathbf{W}$ were adjusted up or down to increase or decrease the spike rate until all neurons spiked at a desired target rate. In this case, we assumed a desired average spike rate of 10 Hz.

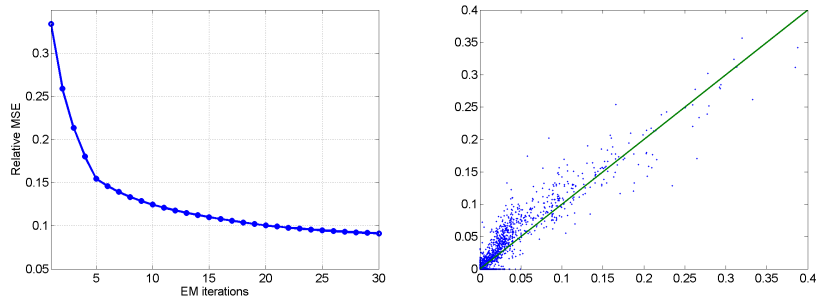

Figure 3: Weight estimation accuracy. Left: Normalized mean-squared error as a function of the iteration number. Right: Scatter plot of the true and estimated weights.

From the parameters in Table 1, we can immediately see the challenges in the estimation. Most importantly, the calcium imaging time constant $\alpha_{CA}$ is set for 500 ms. Since the average neurons spike rate is assumed to be 10 Hz, several spikes will typically appear within a single time constant. Moreover, both the integration time constant and inter-neuron conduction time are much smaller than the

A typical simulation of the network after the stabilization is shown in Fig. 2. Observe that due to the random connectivity, spiking in one neuron can rapidly cause the entire network to fire. This appears as the vertical bright stripes in the lower panel of Fig. 2. This synchronization makes the connectivity detection difficult to detect under temporal blurring of Ca imaging since it is hard to determine which neuron is causing which neuron to fire. Thus, the random matrix is a particularly challenging test case.

The results of the estimation are shown in Fig. 3. The left panel shows the relative mean squared error defined as

$$\text{relative MSE} = \frac{\min_\alpha \sum_{ij} |W_{ij} - \alpha\widehat{W}_{ij}|^2}{\sum_{ij} |W_{ij}|^2}, \tag{16}$$

where $\widehat{W}_{ij}$ is the estimate for the weight $W_{ij}$. The minimization over all $\alpha$ is performed since the method can only estimate the weights up to a constant scaling. The relative MSE is plotted as a function of the EM iteration, where we have performed only a single loopy BP iteration for each EM iteration. We see that after only 30 iterations we obtain a relative MSE of 7% – a number at least comparable to earlier results in [7], but with significantly less computation. The right panel shows a scatter plot of the estimated weights $\widehat{W}_{ij}$ against the true weights $W_{ij}$.

## 5 Conclusions

We have presented a scalable method for inferring connectivity in neural systems from calcium imaging. The method is based on factorizing the systems into scalar dynamical systems with linear connections. Once in this form, state estimation – the key computationally challenging component of the EM estimation – is tractable via approximating message passing methods. The key next step in the work is to test the methods on real data and also provide more comprehensive computational comparisons against current techniques such as [7].

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
