[Reviews · NeurIPS 2014]

Submitted by Assigned_Reviewer_10

Summary:

The authors consider the problem of neural connectivity inference from calcium imaging observations. Their model follows the approach of [7] (Mischenko et. al 2011), a network of spiking neurons coupled together through a matrix of weights that is to be inferred. The important difference is that the spiking neurons are modeled as leaky integrate-and-fire (LIF) neurons instead of generalized linear model. Similarly to [7], they propose an EM approach where at the E step the hidden variables (spikes, membrane potential, calcium concentration) are estimated and the M step is used to estimate the connectivity matrix W and other parameters. The main contribution of the paper is the efficient implementation of the E step using a generalized approximate message passing algorithm (GAMP), which scales linearly both with the number of timesteps and the number of neurons. They also present a sparse probit regression initialization approach for W, that arises naturally from the structure and dynamics of the LIF neurons. The algorithm is applied on synthetic data and the authors claim to achieve performance at least similar to that of previous methods (e.g. [7]) but with significantly less computation.

Comments

- Why are all the nonzero connectivity weights excitatory? The authors cite simplicity but it looks to me that allowing negative weights shouldn't change anything. Why isn't this the case?

- The authors do not address the "common input" problem, according to which connectivity inference is limited from the fact that only subsets of the whole network can be observed at any given point in time. It seems to me that the methods proposed here could be connected with the approach presented in [24] (Keshri et al. 2013) to address this issue. It would be nice to see a discussion about this point, since a potential application of the GAMP algorithm to the case of unobserved units could bring even more substantial computational gains, and address the major limitation of [24].

Quality

The method presented in the paper is technically sound and is based of the GAMP algorithm that has been very successful in the efficient statistical estimation of sparse/structured signals.

Clarity

The paper is at times dense but mostly clearly written, with a lot of useful information presented in the appendix. There are several typos (see below).

Originality & Significance

The approach of the authors is novel and very likely to be of use in future work since it establishes an efficient for approaching this problem. Since the dimensionality of the connectivity problem can be extremely high, computationally efficient methods will be of greater importance. Even though all the tools that are presented here have been presented before, the application is original since the authors consider a LIF based setup that allows them to apply the GAMP algorithm in a non-trivial way.


Typos:

line 21: "methods" possibly missing after "Bayesian inference"
line 107: The second instance of \tilde{v}_i^k should be without the \tilde?
line 107: The bias term b_{IF,i} should probably be moved outside the definition of q_i^k since q is subsequently expressed as q = Ws.
line 109: eq (2) v_i^k < \mu ---> \tilde{v}_i^k < \mu
line 148: \tau_{CA} and \tau{y} in eq. (4) is the same noise variance?
line 183: v^k_{T-1} should be changed to v^k_{N}
line 312: add "to" before a global maxima
line 404: smaller than theā€¦ ??
Summary: A clever and novel application of the generalized approximate message passing algorithm to the problem of connectivity inference in a network of LIF neurons, from calcium imaging observations. The method shows good reconstruction quality, operating in a much faster way than other standard model based approaches.

Submitted by Assigned_Reviewer_18

this manuscript addresses an increasingly important problem in neuroscience: estimating neural connectivity from calcium imaging. the others proposal a "new" model by combining previous models, and a new method to infer connectivity. all is good. some requests for the revision:

-- go over english and math carefully, perhaps a new pair of eyes, there are several typos

-- this sentence in the abstract: "Simulations of the method on realistic neural networks demonstrate good accuracy with computation times that are potentially significantly faster than current approaches based on Markov Chain Monte Carlo methods."

please change 'potentially' to 'actually' if possible. if i recall correctly, our codes for the previous methods are a mess, although they are available. unless yuriy is a co-author here, i'd be surprised if anybody could get it running with relatively little effort. nonetheless, it is the de facto standard, so probably worth explicitly comparing.

-- given the difficulty of making that comparison, a different comparison would be to do what tim did, ie, estimate spiking via foopsi, and then plug the (thresholded) results into the standard GLM package. such an approach, i expect, would be significantly faster than that suggested here. in the simulation proposed here, i expect it might do as well. if so, i imagine there is a regime in which this new method significantly outperforms the simpler strategy. it would be especially exciting if the authors could find such a regime, and then explain (perhaps in the discussion) why the two different methods work better in the two different regimes, and how one might construct a new method that improves on both.

-- with 'scalable' in the title, i would like to see some constants (not just orders of time), and/or wall time. O(N) with a constant of e^10000 is not very scalable :)
Summary: very nice method, important application, could benefit from more explicit comparisons and demonstration of scalability.

Submitted by Assigned_Reviewer_42

The paper "Scalable Inference for Neuronal Connectivity from Calcium
Imaging" proposes and algorithm to estimate the neuron connectivity
from calcium imaging data via a fitting a network of coupled
integrate and fire neurons to the data. The method uses EM and a hybrid of
loopy belief propagation and approximate message passing for
parameter estimation. The algorithm is evaluated on simulated data.

The paper is well written and previous work is sufficiently
acknowledged. The model seems reasonable and the topic is interesting to the NIPS audience.

The weakest spot of the paper are the experiments. The authors
advertise their method as computationally more efficient than other
methods. I can understand that an evaluation on real data as well as a
comprehensive comparison to other methods is maybe a little bit too
much for a limited space NIPS paper. However, a single MSE trace and a
scatter plot with true/estimated weights (without axis labels btw) is
not sufficient. At least one could have expected:

* a plot that shows how the optimization time scales with the number of neurons/connections
* at least either a comparison to other methods or an evaluation on real data

I think the space constraint would have allowed for that. For example,
one could have replaced figure 3, moved table 1 to supplementary, and
possibly dropped figure 2.
Summary: The paper presents and interesting (and possibly computational more efficient) approach for inferring connectivity from calcium imaging data. The weakest part of the paper is the experiments because the method is only evaluated on simulated data, not compared to other methods, and no runtime analysis is presented.
Author Feedback
Author feedback is not available.